# Influence of Carbon Source on Microstructural and Mechanical Properties of High-Performance Reaction-Bonded Silicon Carbide

**DOI:** 10.3390/ma15155250

**Published:** 2022-07-29

**Authors:** Yabin Zhou, Wenhao Sha, Yingying Liu, Yinong Lyu, Yihua Huang

**Affiliations:** 1College of Materials Science and Engineering, Nanjing Tech University, Puzhu South Road No. 30, Nanjing 211816, China; 202061203193@njtech.edu.cn; 2State Key Laboratory of High-Performance Ceramics and Superfine Microstructure, Shanghai Institute of Ceramics, Chinese Academy of Sciences, No. 588, Heshuo Road, Jiading District, Shanghai 201800, China; shawenhao20@mails.ucas.ac.cn (W.S.); liuyingying@student.sic.ac.cn (Y.L.); 3University of Chinese Academy of Sciences, Beijing 100049, China

**Keywords:** carbon black, reaction-bonded silicon carbide, free silicon

## Abstract

Reaction-bonded silicon carbide (RBSC) has become an important structural ceramic with the benefit of being capable of preparing complex-shaped products. In order to fabricate high-performance RBSC, particle gradation of raw SiC combined with slip casting was used to prepare the porous preform before liquid silicon infiltration (LSI). The microstructural and mechanical properties of RBSC were compared by adding different amounts of carbon black (CB) content from 4 wt% to 10 wt%. Two pore structures with submicron and nano pores formed in the preform. As the amounts of carbon black increased, the mechanical properties improved and then suddenly weakened due to residual silicon initiating a nonuniform microstructure. The elastic modulus of the preform with 8 wt%CB after LSI was 389 ± 4 GPa and the flexural strength was 340 ± 17 MPa, which improved by about 150% compared to other rapid prototyping methods and has attractive application prospects.

## 1. Introduction

As an important structural ceramic material, silicon carbide (SiC) is often made into bulletproof armor sheets, mechanical bearings, kiln furniture, burner nozzle, sealing ring and other mechanized tools due to its excellent properties such as excellent hardness, strong wear resistance, fine thermal conductivity, high temperature resistance and corrosion resistance [1,2,3,4]. It is widely used in steel, petrochemical, machinery manufacturing and other fields. Since SiC is a three-dimensional crystal with strong covalent bonding composed of a Si-C tetrahedron and has a diamond-like structure, the diffusion rate during sintering is quite low. At the same time, the oxide layer covering the particle surface acts as a diffusion barrier and hinders diffusion. Therefore, pure SiC is difficult to sinter and compact [5]. To fabricate dense SiC products, hot pressing, pressureless sintering and reactive melt infiltration are usually used. Among these, the hot-pressing sintering process is limited by the availability of equipment. Pressureless sintering causes the shrinkage of products and sintering aids may be needed to help with densifying. Therefore, it is difficult to prepare large-sized and complex-shaped products [6,7]. The reactive melt infiltration (RMI) process can realize the densification of products at a lower temperature (usually above the melting point of Si, 1410 °C), and there is no volume shrinkage change of products during the process. Based on these advantages, the RMI process is the most effective method to prepare carbonized products with a large size and complex shape [8].

Reaction-bonded silicon carbide is fabricated by reactive melt infiltration or liquid silicon infiltration. Reactive melt infiltration is achieved by a capillary force between Si and the porous preform. When the temperature is higher than the melting point of silicon, the capillary action becomes intense between liquid Si and SiC since the wetting angle is less than 90°. Liquid Si continuously infiltrates through the pores of the preform and reacts with the carbon source in the preform to form β-SiC. Β-SiC will grow and connect around the α-SiC particles, while the unresponsive liquid silicon continues to flow in the pores of the preform to discharge bubbles and fill the excess pores [9,10]. There will inevitably be unreacted silicon phase in the ultimate RBSC, which greatly affects its mechanical properties. Particularly after reaching the melting point of silicon, the product performance decreases sharply, which limits the working temperature of RBSC [11]. Therefore, in recent years, researchers have carried out various experiments to investigate how to improve the mechanical properties of RBSC and reduce residual Si.

So far, the preparation processes of RBSC ceramic green bodies mainly include dry-pressing, cold isostatic pressing, gel-casting, additive manufacturing (AM) techniques and slip casting [12,13,14,15]. The disadvantages of dry-pressing and cold isostatic pressing are as follows. It takes a long time to form green bodies and comes with the possibility of cracking. Also, it is particularly difficult to prepare complex-shaped ceramics. Gao et al. [16] First fabricated SiC whisker-reinforced SiC ceramic matrix composites (SiC_W_/SiC) using an RMI process and isotropic whisker preforms were prepared by gel-casting. However, the flexural strength of SiC_W_/SiC is only 218 Mpa and the fracture toughness is 3.11 Mpa m^1/2^. Meyers et al. [17] Fabricated RBSC that had four-point flexural strength of 162 Mpa by laser sintering (LS). Additive manufacturing techniques cannot fabricate large-scale industrial and quality products with high bending strength. Obviously, it is necessary to study how to improve the strength of RBSC in any way. Slip casting is not only simple in process and low-cost, but also suitable for the preparation of large-scale complex ceramic products [18].

Byung koog Jang et al. [19] reported the uni-modal distribution and bi-modal distribution characteristics of SiC raw material powder. The mechanical properties of RBSC made of bi-modal distribution SiC are obviously better than those of uni-modal distribution SiC. Ning song et al. [20] studied the effect of multi-walled carbon nanotubes on the microstructure and mechanical properties of RBSC composites. When the content of multi-walled carbon nanotubes is 10 wt%, the maximum fracture toughness of RBSC is 6.9 Mpa m^1/2^, but its flexural strength deviation is large. Wilhelm et al. [21] first summarized the effect of the original particle size of SiC powder on the mechanical properties of siliconized SiC composites. The average particle sizes used are 12.8, 6.4, 4.5 and 3 μM of SiC powder, and it is observed that the flexural strength increases linearly with the decrease in SiC particle size, and the maximum flexural strength reaches 583 Mpa. However, because the raw material has only one particle size, with the further reduction of SiC particle size, it decreases from 3 to 0.5 μm. The strength decreases gradually. Thus, this study attempts to use SiC with a multi-peak size, and the difference between a large particle size and a small particle size is about 23 times, so as to further improve the elastic modulus of RBSC. 

In order to fabricate RBSC with high performance, the particle size gradation of SiC raw materials and the morphology of carbon source were studied. The slurry required for slip casting is prepared by using reasonable gradation and adding a small amount of carbon black. The relationship between microstructure and properties of RBSC was analyzed.

## 2. Experimental Procedure

First, SiC-1 and SiC-2 are both α-SiC with a hexagonal crystal structure. SiC raw materials were particle-graded, different contents of carbon black were added, mixed with distilled water for 6 h, and then vacuum debubbled. The viscosity of the slurry was measured using a rheometer. The slurry lost water and formed the green body in the gypsum mold which was customized for slip casting. The green body contained SiC with the mass ratio of large particles to small particles of 7:3. Second, vacuum debonding was carried out in the vacuum furnace to obtain the porous preform with a certain strength. The apparent porosity and relative density of the porous preform were measured by the Archimedes’ method. The microstructure of the preform was observed by using a scanning electron microscope. Third, the preform was put in the graphite crucible which was protected with a boron nitride lubricating spray in case of a reaction between Si and the crucibles. The appropriate silicon was weighed to wrap the preform. The mass ratio of Si particles to the preform was 1:1. The porous preform experienced liquid silicon infiltration at 1550 °C for 30 min and formed the RBSC samples. Finally, RBSC samples were processed into 3 × 4 × 36 mm test strips. Then, their mechanical properties were tested using a universal testing machine. The composition of the sample was analyzed with a X-ray diffraction scanner. The polished surface and fracture surface of the sample were observed using a scanning electron microscope. The apparent porosity and relative density of samples were measured by the Archimedes’ method. The content of free-Si in the sample was analyzed by chemical corrosion.

## 3. Results and Discussion 

### 3.1. Characteristics of Raw Materials and Porous Preforms

Figure 1 shows the distribution function of raw material particle size used to prepare RBSC. Figure 1a shows that the two types of SiC powders both have uni-modal distribution. The D_50_ particle size of SiC-1 was 88 μm and SiC-2 was 3.7 μm. Figure 1b CB presents a multi-modal distribution with the particle size on three gradients. The D_50_ particle size of CB was 0.6 μm and D_10_ was 200 nm, which may have caused the formation of hierarchical porous structures in the green body [22]. The graphite had a wide uni-modal distribution and D_50_ was 1.2 μm, as shown in Figure 1c. According to Figure 2a,b, the SiC particles were mostly massive or flaky and a small amount of wrinkled impurities on the SiC-2 surface may have required pickling to remove. The carbon black was in the shape of microspheres, as shown in Figure 2c. With the decrease in particle radius, the surface energy and surface tension increased sharply. The microspheres agglomerated to form larger spheres, which could explain the multi-modal structure in Figure 1b. It could cause trouble when preparing slurries with better dispersivity and could create partial grain growth [22]. Figure 2d reflects a layered structure of multi-layer graphite flakes. The flaky structure has disadvantages in the filling and accumulation of particles. It can hence be inferred that plasticity and slipperiness are inferior to spherical carbon black, when the graphite is used to prepare fluids.

The fluidity of the slurries with different carbon sources is shown in Figure 3. At the beginning, the viscosity of 10 wt% graphite slurry reached 12 Pa·S in Figure 3a and the required shear stress was 20 Pa in Figure 3b, while the 10 wt%CB slurry was much less than the former. The fluidity of slurry with carbon black as its carbon source was obviously superior to graphite. Flake and fibrous fillers have a great influence on the viscosity of the filling system, while granular fillers have little influence. Among the various shapes of granular powder, spherical and quasi spherical powder have better fluidity, which can effectively reduce the influence of filler on the viscosity of the filling system, so as to achieve a higher filling amount. Therefore, graphite was not suitable as a carbon source to prepare the RBSC green body in the process of slip casting. There is an urgent need for excellent dispersant or powder modification technology to optimize the preparation of slurry.

Figure 4 shows the pore size distribution of preforms with different CB content. The pore size distribution of all preforms was wide, and most of the pore sizes were concentrated between 100 nm and 1 μm. A small part of mesopores below 50 nm were formed between carbon particles due to electrostatic attraction. The peak pore size of green billet with 4 wt%CB was the largest. With the increase in carbon content, the peak position of pore size moved to smaller pores, because the agglomeration and adhesion of nano carbon black were more obvious. In particular, compared with the other three, the green body with 6 wt%CB did not have a few large pores of 10 μm. The existence of macropores may have allowed some silicon that had not reacted with carbon to remain in the channel to form residual silicon. The pore structure of the preform graded by SiC particles of different sizes presented a dual pore structure with sub-micron pores as the main and nano pores as the auxiliary.

Figure 5 shows the SEM micrographs of preforms with different carbon contents. The gradation of raw material particles has a significant effect on the fluidity of the suspended slurry. According to the SEM of the green body after debonding, it can be seen that the micron SiC particles were in the shape of long edges, stacked into a frame and forming large cavities. Submicron SiC particles were in flakes or blocks, which were connected to fill the large holes between frames and formed channels. Nano carbon black is elliptical or spherical, with a large specific surface area and a large surface energy between particles. Therefore, the energy state tended to be unstable, and it was easy to agglomerate dozens or hundreds of carbon black. Carbon black spheres adhered around the SiC surface to further fill the channels between SiC and formed sufficient small pores. A few smaller pores would also be formed between carbon black particles, which can be seen from the pore size distribution diagram. Experiments show that reasonable gradation improves the solid content of slurry and ensures good fluidity. The micrograph of C sample scheme shows better close-packing. This grading effectively increases the friction between particles and prevents the slip phenomenon.

Figure 6 shows the bulk density and porosity of preforms depending on the CB content. The bulk density of the preform with 10 wt%CB was 2.23 ± 0.01 g cm^−3^ and the number reduced with the decrease in CB. Higher density can effectively prevent cracking during debonding, because the connection between SiC and C becomes tight and reduces looseness. The porosity of the green body was the largest at 4 wt%CB, and decreases with the increase in carbon black content. The large porosity of porous preform was conducive to the infiltration of Si, but it will also produce free-Si. When excess carbon was added, the porosity decreased and the agglomeration of carbon raw materials made it difficult to disperse evenly. The residual carbon may also be produced during LSI. The carbon preform with low porosity and narrow pore size distribution may have low free-Si and residual carbon [23]. Therefore, the 8 wt%CB preform with a porosity of 24.68 ± 0.23% may exhibit less free-Si compared to other preforms after LSI. It is important to choose a suitable carbon content to promote densification of RBSC and reduce the free-Si.

### 3.2. Microstructure Analysis of Preforms after LSI

All specimens consisted of two main phases of Si and SiC and no carbon peak was detected according to Figure 7. Carbon black reacted with Si to form second phase SiC which was composed of low-temperature stable phase β-SiC and high-temperature α-SiC. Since the graded raw material contained large particles of SiC, high-temperature stable α-SiC also existed in the RBSC. Accordingly, the predominance of any of these phases was not established. Compared to other RBSCs, the 8 wt%CB specimen had the lowest fraction of Si phase derived from the weakest peak intensity of Si, as shown in Figure 7. The low content of Si corresponded to the reduced porosity in Figure 6. Generally speaking, free Si increased due to the uneven density of the green body during slip casting. The excess carbon particles adhered to each other to form agglomerates, the pore size became smaller, and the newly generated β-SiC phase of Si and C blocked the silicon infiltration channel. Since XRD analysis cannot analyze trace particles and heterogeneous particles, the microstructure of RBSC was further explored in detail through the SEM images of the samples.

Figure 8 shows the SEM micrographs of the polished surface of RBSCs. Dark black particles are original α-SiC particles with two particle sizes. The white area is free-Si. The fine gray particles dispersed around the white area are β-SiC formed by the reaction of Si and C. All samples formed β-SiC and free Si. The microstructure of RBSC with 6 wt%CB is more uniform in Figure 8b. With the increase in carbon black content, an obvious Si band region was formed between SiC particles. The Si phase diffused to the grain boundary, which destroyed the stability of the grain boundary and reduced the energy of the grain boundary. Free Si was produced by weak interface bonding relative to the SiC crystal. Cracks first formed around the free Si phase when RBSC was subjected to external forces. Excess carbon black could result in the volume expansion and the exothermic reaction of carbon to weaken strength during LSI [24]. This explained the large channels that remained due to the mutual adsorption of nano carbon black into clusters. Free-Si will result in the instability of RBSC materials at high temperature.

The SEM micrographs of the fracture surface of RBSC is presented in to Figure 9. The main fracture mode of all samples was intergranular fracture due to the presence of smooth black and white block grain faces shown in Figure 9. RBSC prepared by reasonable grading was very dense. No obvious pores or cracks were observed in Figure 9. More transgranular fractures are found in Figure 9d, which appeared as undulating and uneven surfaces. Since transgranular fracture needs to consume more energy, the sample with 10 wt% carbon could show poor mechanical properties. The existence of the Si band region destroyed the stability of SiC connection of large and small particles and led to brittle fracture of the material.

### 3.3. Performance Analysis of RBSC Samples

Figure 10 shows the flexural strength and elastic modulus as a function of carbon black content. The elastic modulus reflects the bonding strength of chemical bonds between atoms. The elastic modulus of RBSC was the largest when 8 wt%CB was used. The flexural strength increased rapidly at the beginning with the content of carbon black. 4 wt%CB content was not sufficient to produce enough second-phase SiC. When the proportion of carbon black was further increased to 10 wt%, the flexural strength decreased. The non-metallicity of C was stronger than that of Si. The diffusion rate of C was lower than Si. The diffusion distance of Si-C reaction increased [25]. When the carbon black content was high, it was difficult to disperse the carbon black evenly into the matrix. The locally accumulated residual Si or carbon made it easy to destroy the uniform microstructure of sintered body. The uneven microstructure affected the mechanical properties of the material. Therefore, the sharp decrease in elastic modulus by 35% was attributed to the uneven pore channel caused by the aggregation of carbon particles during slip casting. This not only hindered the uniform structure of RBSC composites, but also led to the formation of residual silicon.

The change in free-Si was irregular in RBSC as shown in Figure 11. The sample with 10 wt%CB contained the most free-Si of 23.21 wt%. Compared with the lowest 15.92 wt% free-Si, it increased by 46%. This corresponded to the results inferred from XRD in Figure 7. The presence of the most free-Si could also explain the sudden drop in the elastic modulus shown in Figure 10. When the reaction of the SiC phase was completed, a small part of the silicon phase diffused to the grain boundary, which destroyed the stability of the grain boundary. Grain boundary energy decreased. When the sintered body encountered external pressure, the microcrack first occurred at the grain boundary. It propagated along the grain boundary with low strength, which showed intergranular fracture. This eventually led to the destruction of the RBSC structure.

Figure 12 shows the relative density and apparent porosity of RBSCs. With the increase in carbon black content, the porosity decreased gradually. The apparent porosity of RBSC with 10 wt%CB was 0.29%. After the carbon black gathered, it left a large channel. This was conducive to the rapid infiltration of Si and the discharge of bubbles. However, the RBSC density with 10 wt%CB decreased sharply due to the existence of 23.21 wt% residual Si in the channel. The density of Si was less than that of SiC. RBSC with 4 wt%CB did not generate much β-SiC due to the lack of carbon black. This led to its low density. RBSC with 8 wt%CB had a high bulk density (2.96 ± 0.04 g cm^−3^) and low apparent porosity. Its high density corresponded to the high elastic modulus. Reasonable carbon black content reconciled the content of β-SiC and free Si. This grading scheme effectively obtained the uniformly arranged SiC microstructure. Thus, the compactness and excellent mechanical properties of RBSC were improved.

## 4. Conclusions

Reaction-bonded SiC with high flexural strength (340 ± 17 Mpa, 6 wt%CB) and an elastic modulus (389 ± 4 Gpa, 8 wt%CB) was successfully obtained by liquid-phase siliconization at 1550 °C vacuum for 30 min. The design of CB distribution could reduce the content of free silicon and effectively adjust the residual stress caused by thermal stress. The fluidity of the slurry prepared by 10 wt%CB was better than 10 wt% graphite. The SiC preform which was graded by different particle sizes showed a dual pore structure of sub-micron and nano pore. The microstructure of the preform with 6 wt%CB showed the most uniform pore-size distribution framework. The minimum free-Si content of 8 wt%CB was 15.92 wt%. After further increasing the carbon black content, the mechanical properties decreased significantly. The microstructure of SiC skeleton was arranged unevenly. The content of free-Si increased continuously, which was manifested by the formation of the Si band region. The main fracture mode of SiC with higher strength was an intergranular fracture. The addition of large particles hindered crack propagation. Free-Si destroyed the stability of the grain boundary and reduced the energy of the grain boundary. The results showed that using reasonable gradation and a carbon source is an effective method to improve the mechanical properties of reaction-bonded SiC. This method is conducive to the fabrication of RBSC to make products with a large size and complex shape, which has attractive prospects for industrialization.

## Figures and Tables

**Figure 1 materials-15-05250-f001:**
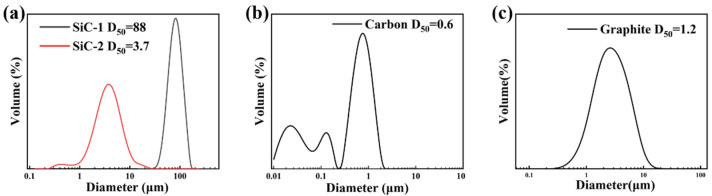
Particle size distribution diagram of the raw material. (**a**) SiC-1 and SiC-2; (**b**) carbon black; (**c**) graphite.

**Figure 2 materials-15-05250-f002:**
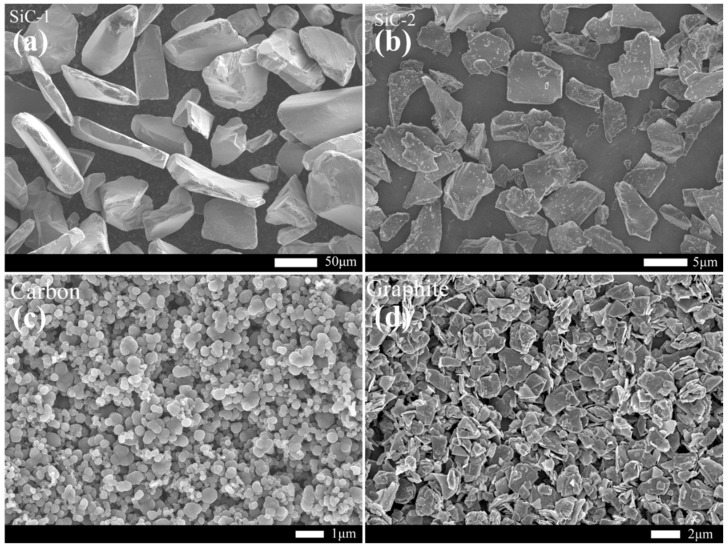
SEM micrographs of raw material powders.

**Figure 3 materials-15-05250-f003:**
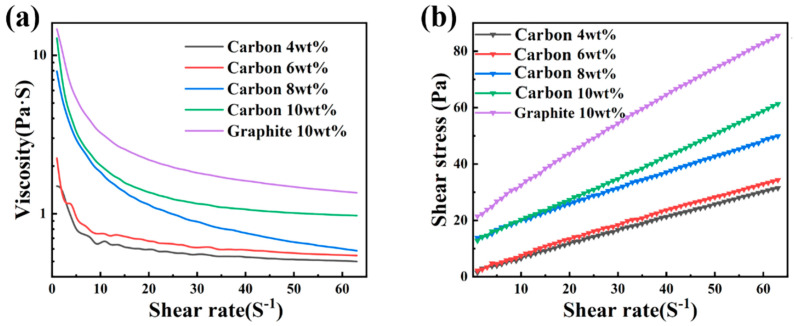
Rheological function of slurries prepared from different carbon sources (a-viscosity curve and b-shear stress curve).

**Figure 4 materials-15-05250-f004:**
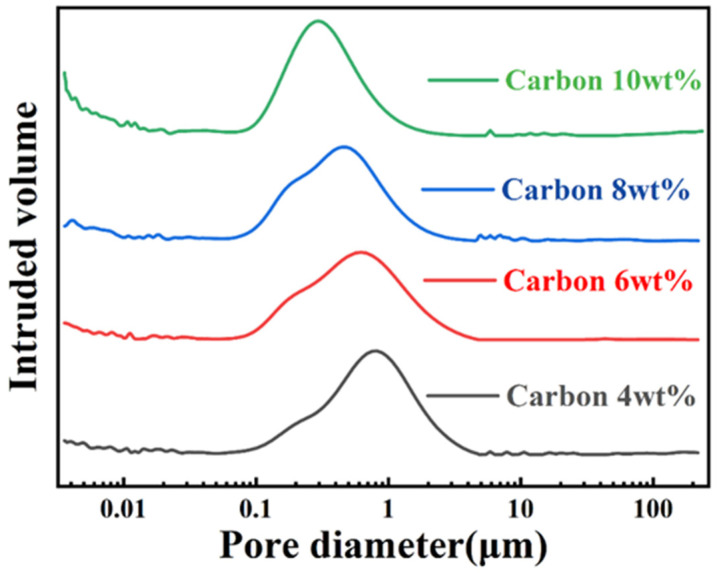
The pore size distribution of preforms with different CB content.

**Figure 5 materials-15-05250-f005:**
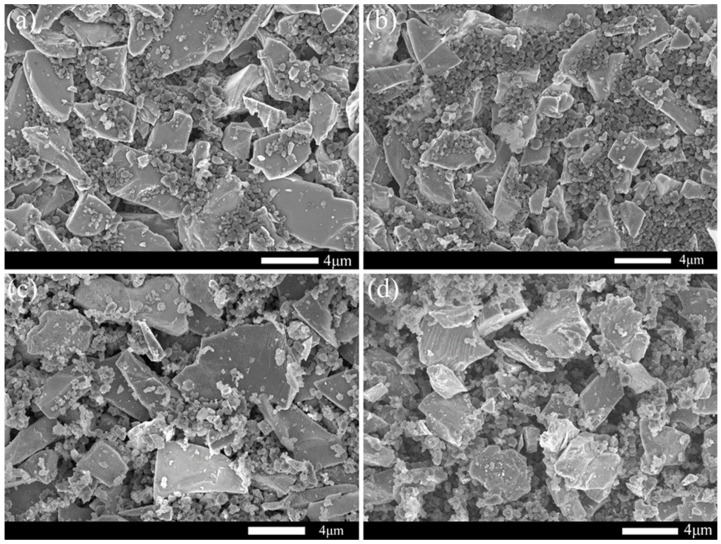
SEM micrographs of preforms (**a**) 4 wt%CB, (**b**) 6 wt%CB, (**c**) 8 wt%CB and (**d**) 10 wt%CB.

**Figure 6 materials-15-05250-f006:**
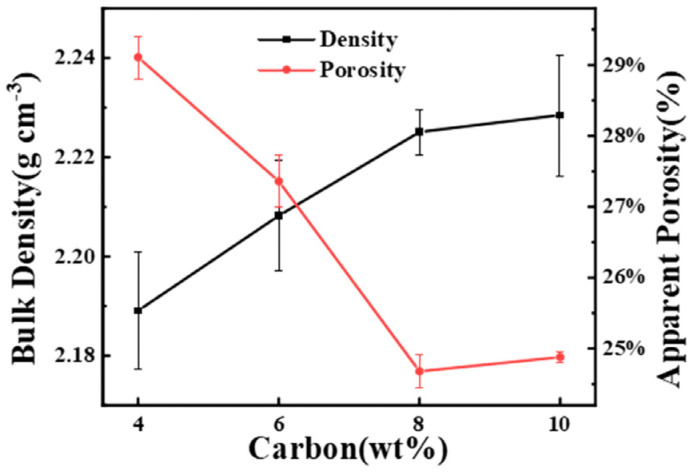
Bulk density and porosity of preforms with different CB contents.

**Figure 7 materials-15-05250-f007:**
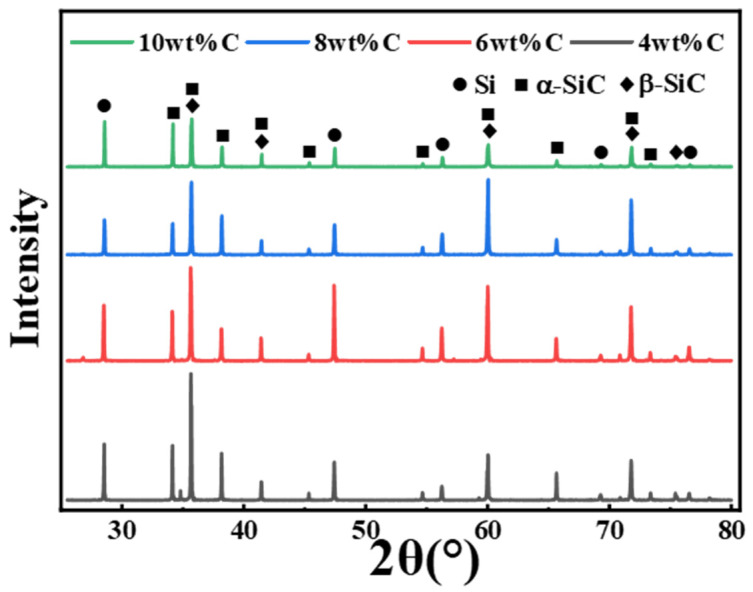
X-ray diffraction pattern of the RBSC depending on different carbon content.

**Figure 8 materials-15-05250-f008:**
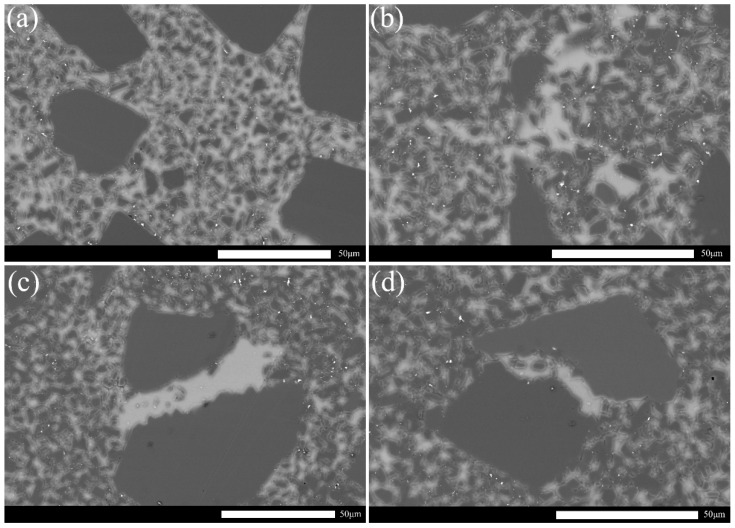
SEM micrographs of the polished surface of the RBSC with different carbon content (**a**) 4 wt%CB, (**b**) 6 wt%CB, (**c**) 8 wt%CB and (**d**) 10 wt%CB.

**Figure 9 materials-15-05250-f009:**
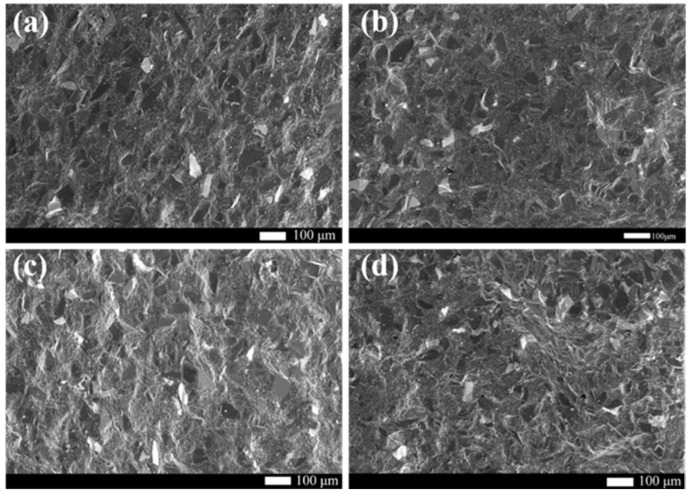
SEM micrographs of the fracture surface of the RBSC with different carbon content (**a**) 4 wt%CB, (**b**) 6 wt%CB, (**c**) 8 wt%CB and (**d**) 10 wt%CB.

**Figure 10 materials-15-05250-f010:**
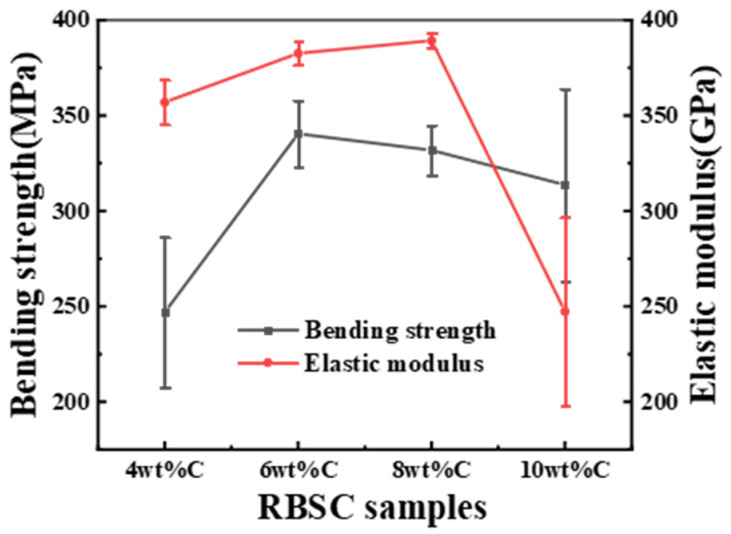
Flexural strength and elastic modulus as a function of RBSCs with different CB content.

**Figure 11 materials-15-05250-f011:**
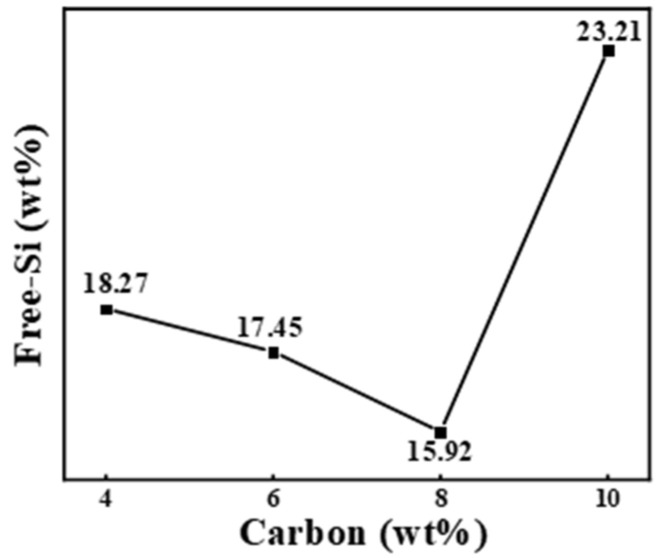
Diagram of free-Si as a function of CB content.

**Figure 12 materials-15-05250-f012:**
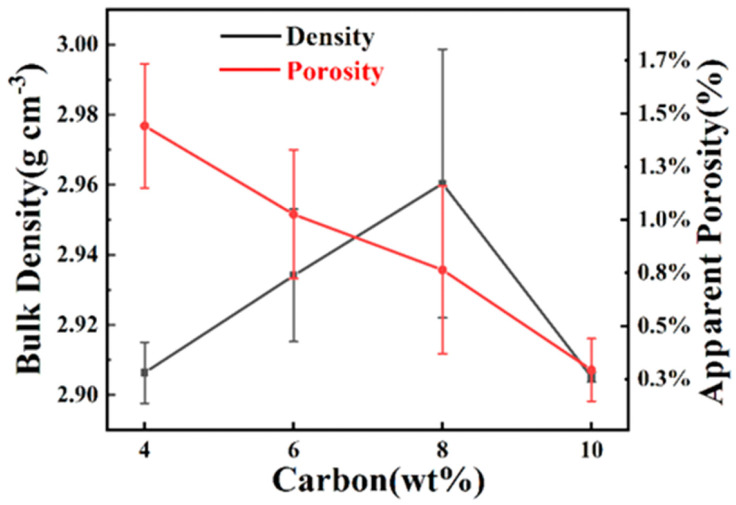
Bulk density and porosity of RBSCs with different carbon content.

## Data Availability

Not applicable.

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
