# Peer review of "Influence of Carbon Source on Microstructural and Mechanical Properties of High-Performance Reaction-Bonded Silicon Carbide"

_materials, 2022, doi:10.3390/ma15155250_

Round 1

Reviewer 1 Report

Line 46: Explain sentance better.

Explain little more RMI proces in introduction and compare with RBSC.

Line 61: Gao et al,.. explain sentance?

Line 88: techical things.

In part 2 Experiment, can you explain Archimede's methods? What is BN in line 98?

Figure 1c, mark on graph D50, so can be all three uniform.

Figure 8 sholud be more clearly.

Some more explanation should be added in the conclusion to make it more convincing.Based on what was written, it is unclear whether is obtanied material has acceptable mechanical properties or not.

Author Response

Dear Editors and Reviewers:

       Thank you very much for your comments on our manuscript entitled “Influence of Carbon Source on Microstructural and Mechanical properties of High-Performance Reaction-bonded Silicon Carbide”. Those comments are all valuable and very helpful for revising and improving our paper, as well as the important guiding significance to our researches. We have studied comments carefully and have made corrections which we hope to meet with approval. Revised portions are marked in red on the paper. The main corrections in the paper are addressed point by point below:

Reviewer #1: Comment 1: Line 46: Explain sentance better.

Response: Thank you for your suggestion. We have corrected it in line 46.

Comment 2: Explain little more RMI proces in introduction and compare with RBSC.

Response: Thank you for your suggestion. We have added explanations about RMI. Reaction-bonded silicon carbide(RBSC) is fabricated by reactive melt infiltration(RMI) or liquid silicon infiltration(LSI). RBSC is a kind of SiC ceramic product. RMI and LSI are sintering technologys. The sintering process is the same, but it has different names.

Comment 3: Line 61: Gao et al,.. explain sentance?

Response: Thank you for your suggestion. We have added the explanation.

Comment 4: Line 88: techical things. We have corrected it in line 88.

Response: Thank you for your suggestion.

Comment 5: In part 2 Experiment, can you explain Archimede's methods? What is BN in line 98?

Response: Thank you for your suggestion. Archimede's methods: ρ= m1/(m2-m3) *ρwater, Apparent porosity=(m2-m1)/(m2-m3), Where m1 is the dry weight; m2 is the wet weight; m3 is the mass of the sample floating in the water.

BN is boron nitride lubricating spray in case of reaction between Si and crucibles.

Comment 6: Figure 1c, mark on graph D50, so can be all three uniform.

Response: Thank you for your suggestion. We have corrected it in Figure 1c.

Comment 7: Figure 8 sholud be more clearly.

Response: Thank you for your suggestion. We have added the analyses in the paper.

Comment 8: Some more explanation should be added in the conclusion to make it more convincing.Based on what was written, it is unclear whether is obtanied material has acceptable mechanical properties or not.

Response: Thank you for your suggestion. We have added the explanations in paper. The mechanical properties have exceeded the best report in the other literatures.

Reviewer 2 Report

The manuscript “Influence of Carbon Source on Microstructural and Mechanical properties of High-Performance Reaction-bonded Silicon Carbide” can be published after revision. The reviewer’s comments are listed below.

Please correct the list of authors.

Abstract and conclusions. The values of flexural strength of 340.49±17.49 MPa and elastic modulus of 389.22±3.93 GPa are meaningless. Please make the following correction, 340±17 MPa and 389±4.

Please give a brief description of the reactive melt infiltration process.

It seems that the text presented in lines37 – 39 is repeated in lines 57 – 61. Please make correction.

Line 69. “Byung koog Jang et al. reported the unimodal distribution and bimodal distribution characteristics of raw material powder.” Please indicate what raw material you mean.

Line 73. “When the content is 10wt%,…” Please explain which content you mean.

Line 89. Please indicate the structure of SiC.

Line 105. Please explain the method of chemical corrosion and the error of free-Si estimation.

Figure 1. Please check if you mean “D50 particle size distribution diagram” or particle size distribution diagram. In Fig. 1(b), the particle size distribution diagram is presented.

Line 184. Please give the error of the porosity estimation.

Line 190. The authors state that “Carbon black reacted with Si to form the low temperature stable phase beta-SiC in all samples. Since the graded raw material contained large particles of SiC, high temperature stable alpha-SiC also existed in the RBSC.” From the XRD pattern, we see the coexistence of both crystalline phases in all samples; the presence of alpha phase is manifested in all samples, the presence of the beta phase is seen only by the peak at 2theta about 75.7o. The XRD pattern does not allow to estimate the relative fractions of both phases. So I would rather say that carbon black reacted with Si to form both the beta and the alpha forms of SiC. The predominance of any of these phases is not established.

Please provide the XRD standard card numbers that you used for the estimation of the crystalline phases.

Line 192. I cannot agree with the statement that “It can be seen that in the XRD analysis results, the relative intensity of residual Si peak increases with the increase of carbon black content.” The highest intensity of diffraction peaks of Si is found in the sample with 6% C, see Fig. 7.

Please give your comments on the redistribution of intensities of diffraction peaks of SiC in samples with different content of Si, please especially pay attention on the XRD pattern of the sample with 10% Si.

Line 237. “Therefore, the sharp decrease of elastic modulus by 35% is attributed to the aggregation of carbon particles.” Please explain why we do not see the carbon the XRD patterns and the SEM images.

Lines 243, 259, 274. Please explain how you determined the content of free-Si in your samples. Please provide the error of its estimation.

Please prepare all references in the same style.

The grammar correction is required. Below I list some sentences that require correction.

I cannot understand the meaning of the sentence, line 17. “The carbon black(CB) content of RBSC was compared horizontally from 4wt% to 10wt%.”

Line 18. “A dual pore structure with submicron and nano pores formed in the preform.”

Line 19. “As the carbon black increased, mechanical properties increased…”

I cannot understand the meaning of the sentence, line 46 “The principle of RBSC is to use the wetting angle between liquid Si and SiC with high reactivity at high temperature is less than 90°, and the capillary action is obvious.”

Line 59. “The dry pressing and cold isostatic pressing process has the disadvantages of being time-consuming, easy to crack, and unable to prepare ceramics with complex shapes.”

Line 88. Please correct typos.

Line 114. “Fig1(a) two types of SiC powders are all unimodal distribution.”

I cannot understand the meaning of the sentence, line 115. “Fig1(b) CB presents a multimodal distribution.”

Line 123. “It may cause difficulties in preparing well dispersed slurries and create localize grain growth[22].”

Line 126. “particles So it can be inferred that plasticity and slipperiness of are inferior to spherical carbon black, when the graphite is used to prepare fluids.”

Line 131. “At the beginning, the viscosity of graphite slurry reached 12 Pa·S and the required shear stress was 20 Pa, while the carbon black slurry was much less than the former”

Line 270. “The slurry prepared by carbon black has better fluidity than graphite”

Author Response

Dear Editors and Reviewers:

       Thank you very much for your comments on our manuscript entitled “Influence of Carbon Source on Microstructural and Mechanical properties of High-Performance Reaction-bonded Silicon Carbide”. Those comments are all valuable and very helpful for revising and improving our paper, as well as the important guiding significance to our researches. We have studied comments carefully and have made corrections which we hope to meet with approval. Revised portions are marked in red on the paper. The main corrections in the paper are addressed point by point below:

Reviewer #2:Reviewer's General Comments

The manuscript “Influence of Carbon Source on Microstructural and Mechanical properties of High-Performance Reaction-bonded Silicon Carbide” can be published after revision. The reviewer’s comments are listed below.

Response: Thank you for your comments on this article. We have further revised the contents of the manuscript. We sincerely thank you for your many improvements in our writing.

Comment 1: Please correct the list of authors.

Response: Thank you for your suggestion. We have corrected it.

Comment 2: Abstract and conclusions. The values of flexural strength of 340.49±17.49 MPa and elastic modulus of 389.22±3.93 GPa are meaningless. Please make the following correction, 340±17 MPa and 389±4

Response: Thank you for your suggestion. We have corrected it in the abstract and conclusions.

Comment 3: Please give a brief description of the reactive melt infiltration process.

Response: Thank you for your comments. We have provided a description in the paper. Reactive melt infiltration is achieved by capillary force between Si and porous preform. When the temperature is higher than the melting point of silicon, the capillary action becomes intense between liquid Si and SiC since the wetting angle is less than 90°.

Comment 4: It seems that the text presented in lines37 – 39 is repeated in lines 57 – 61. Please make correction.

Response: Thank you for your comments. We are sorry for the confusion caused to you. The text presented in lines37-39 is comparison of sintering methods of SiC while the preparation processes of SiC powder into green bodies is compared in lines 57-61.

Comment 5: Line 69. “Byung koog Jang et al. reported the unimodal distribution and bimodal distribution characteristics of raw material powder.” Please indicate what raw material you mean.

Response: Thank you for your comments. We have added SiC raw material in the paper.

Comment 6: Line 73. “When the content is 10wt%,…” Please explain which content you mean.

Response: Thank you for your suggestions. We are sorry we forgot to write the content of multi-walled carbon nanotubes

Comment 7: Line 89. Please indicate the structure of SiC.

Response: Thank you for your advice.We have added the structure of SiC.

Comment 8: Line 105. Please explain the method of chemical corrosion and the error of free-Si estimation.

Response: Thank you for your comments. Soaking RBSC powders in boiling NaOH can make free-Si react to release hydrogen, and the content of free-Si can be indirectly determined by measuring the volume of hydrogen. The relative standard deviation is 0.03%.

Comment 9: Figure 1. Please check if you mean “D50 particle size distribution diagram” or particle size distribution diagram. In Fig. 1(b), the particle size distribution diagram is presented.

Response: Thank you for your comments. We mean particle size distribution diagram of Figure 1.

Comment 10: Line 184. Please give the error of the porosity estimation.

Response: Thank you for your advice. We have added the error of the porosity estimation.

Comment 11: Line 190. The authors state that “Carbon black reacted with Si to form the low temperature stable phase beta-SiC in all samples. Since the graded raw material contained large particles of SiC, high temperature stable alpha-SiC also existed in the RBSC.” From the XRD pattern, we see the coexistence of both crystalline phases in all samples; the presence of alpha phase is manifested in all samples, the presence of the beta phase is seen only by the peak at 2theta about 75.7o. The XRD pattern does not allow to estimate the relative fractions of both phases. So I would rather say that carbon black reacted with Si to form both the beta and the alpha forms of SiC. The predominance of any of these phases is not established.

Response: Thank you for your comments. We think your statement is more accurate. We have made changes in the text.

Comment 12: Please provide the XRD standard card numbers that you used for the estimation of the crystalline phases.

Response: Thank you for your comments. PDF Number: 29-1131, 73-1663 and 89-5012.

Comment 13: Line 192. I cannot agree with the statement that “It can be seen that in the XRD analysis results, the relative intensity of residual Si peak increases with the increase of carbon black content.” The highest intensity of diffraction peaks of Si is found in the sample with 6% C, see Fig. 7.

Response: Thank you for your suggestions. We have given new comments about Figure7.

Comment 14: Please give your comments on the redistribution of intensities of diffraction peaks of SiC in samples with different content of Si, please especially pay attention on the XRD pattern of the sample with 10% Si.

Response: Thank you for your suggestions. We have given new comments about Figure7.

Comment 15: Line 237. “Therefore, the sharp decrease of elastic modulus by 35% is attributed to the aggregation of carbon particles.” Please explain why we do not see the carbon the XRD patterns and the SEM images.

Response: Thank you for your comments. We are sorry for the confusion caused to you due to vague statement. We have changed it.

Comment 16: Lines 243, 259, 274. Please explain how you determined the content of free-Si in your samples. Please provide the error of its estimation.

Response: Thank you for your questions. Soaking RBSC powders in boiling NaOH can make free-Si react to release hydrogen, and the content of free-Si can be indirectly determined by measuring the volume of hydrogen. The relative standard deviation is 0.03%.

Comment 17: Please prepare all references in the same style

Response: Thank you for your suggestions. We have made changes in the manuscript.

Comment 18: The grammar correction is required. Below I list some sentences that require correction.

Response: Thank you for your suggestions. We have made changes in the manuscript.

Comment 19: I cannot understand the meaning of the sentence, line 17. “The carbon black(CB) content of RBSC was compared horizontally from 4wt% to 10wt%.”

Response: Thank you for your suggestions. We have made changes in the manuscript.

Comment 20: Line 18. “A dual pore structure with submicron and nano pores formed in the preform.”

Response: Thank you for your suggestions. We have made changes in the manuscript.

Comment 21: Line 19. “As the carbon black increased, mechanical properties increased…”

Response: Thank you for your suggestions. We have made changes in the manuscript.

Comment 22: I cannot understand the meaning of the sentence, line 46 “The principle of RBSC is to use the wetting angle between liquid Si and SiC with high reactivity at high temperature is less than 90°, and the capillary action is obvious.”

Response: Thank you for your suggestions. We have made changes in the manuscript.

Comment 23: Line 59. “The dry pressing and cold isostatic pressing process has the disadvantages of being time-consuming, easy to crack, and unable to prepare ceramics with complex shapes.”

Response: Thank you for your suggestions. We have made changes in the manuscript.

Comment 24: Line 88. Please correct typos.

Response: Thank you for your suggestions.We have made changes in the manuscript.

Comment 25: Line 114. “Fig1(a) two types of SiC powders are all unimodal distribution.”

Response: Thank you for your suggestions. We have made changes in the manuscript.

Comment 26: I cannot understand the meaning of the sentence, line 115. “Fig1(b) CB presents a multimodal distribution.”

Response: Thank you for your suggestions. We have made changes in the manuscript.

Comment 27: Line 123. “It may cause difficulties in preparing well dispersed slurries and create localize grain growth[22].”

Response: Thank you for your suggestions. We have made changes in the manuscript.

Comment 28: Line 126. “particles So it can be inferred that plasticity and slipperiness of are inferior to spherical carbon black, when the graphite is used to prepare fluids.”

Response: Thank you for your suggestions. We have made changes in the manuscript.

Comment 29: Line 131. “At the beginning, the viscosity of graphite slurry reached 12 Pa·S and the required shear stress was 20 Pa, while the carbon black slurry was much less than the former”

Response: Thank you for your suggestions. We have made changes in the manuscript.

Comment 30: Line 270. “The slurry prepared by carbon black has better fluidity than graphite”

Response: Thank you for your suggestions. We have made changes in the manuscript.

Round 2

Reviewer 2 Report

The manuscript can be published.

Author Response

We are very honored to have your recognition of this work. Thank you again for your valuable comments, which have played a great role in improving the quality of the article.